# Improved CNN Prediction Based Reversible Data Hiding for Images

**DOI:** 10.3390/e27020159

**Published:** 2025-02-03

**Authors:** Yingqiang Qiu, Wanli Peng, Xiaodan Lin

**Affiliations:** 1College of Information Science & Engineering, Huaqiao University, Xiamen 361021, China; xd_lin@hqu.edu.cn; 2School of Computer Science, Fudan University, Shanghai 200433, China; pengwanli@fudan.edu.cn

**Keywords:** CNN, multitasking, reversible data hiding, histogram shifting

## Abstract

This paper proposes a reversible data hiding (RDH) scheme for images with an improved convolutional neural network (CNN) predictor (ICNNP) that consists of three modules for feature extraction, pixel prediction, and complexity prediction, respectively. Due to predicting the complexity of each pixel with the ICNNP during the embedding process, the proposed scheme can achieve superior performance compared to a CNNP-based scheme. Specifically, an input image is first split into two sub-images, i.e., a “Circle” sub-image and a “Square” sub-image. Meanwhile, each sub-image is applied to predict another one with the ICNNP. Then, the prediction errors of pixels are sorted based on the predicted pixel complexities. In light of this, some sorted prediction errors with less complexity are selected to be efficiently applied for low-distortion data embedding with a traditional histogram-shifting technique. Experimental results show that the proposed ICNNP can achieve better rate-distortion performance than the CNNP, demonstrating its effectiveness.

## 1. Introduction

Reversible data hiding (RDH) can extract embedded data correctly and recover cover media without any loss [1,2]. Due to these traits, RDH has become a research focus in the community of information hiding and has been widely applied in several realistic scenarios, including medical, military, and law forensics. Based on the domain hiding additional data, RDH can be categorized into two main categories: spatial domain-based RDH [3,4,5,6,7,8,9,10,11,12,13,14,15,16,17,18,19,20,21,22,23,24,25,26] and JPEG domain-based RDH [27,28,29,30,31,32]. Spatial domain-based RDH generally utilizes three classic techniques and their extensions, i.e., lossless compression (LC) [3,4,5], difference expansion (DE) [6,7,8,9,10,11,12,13,14,15,16,17,18], and histogram shifting (HS) [19,20,21,22,23,24,25,26]. In contrast, JPEG domain-based RDH is mainly based on DCT coefficient modification [27,28,29,30] or Huffman table modification [31,32].

Currently, in the RDH field, pixel prediction has become a key point, which dramatically affects the performance of RDH algorithms [15]. Traditional predictors include the median edge direction predictor (MEDP) [7], interpolation predictor [8], gradient-adjusted predictor (GAP) [9], pixel value ordering (PVO) predictor [10,13,25], linear predictor [11], rhombus predictor [20,21,22,23], ridge regression predictor [26], etc. Although these predictors have achieved visual improvement, there is still a notable weakness, which is that few adjacent pixels are applied for pixel prediction [15]. If more adjacent pixels serve as reference pixels, a higher prediction performance can be achieved. Due to its strong capabilities of fusing different receptive fields and whole optimization, a convolutional neural network (CNN) can be constructed and trained to precisely predict pixels by building a nonlinear mapping for pixel prediction. In light of this, Luo et al. [14] presented a CNN-based stereo image RDH scheme leveraging the correlations between right and left views. Hu et al. [15] proposed a CNN predictor (CNNP)-based RDH scheme, where a grayscale image is split into two sub-images, and each one is predicted with another one alternatively using the CNNP. After that, Hu et al. [16] divided a cover image into four parts, and each part was predicted with the other three parts in turn using a CNNP for a better prediction performance. Furthermore, superior performance is attained through prediction-error-ordering (PEO)-based adaptive embedding. Overall, the prediction performance of CNN predictors can be better than that of traditional predictors. In [17], Yang et al. introduced a novel RDH approach that strategically segments a cover image into four distinct regions. Within each region, each pixel is then predicted using the surrounding eight neighbor pixels through a custom CNN predictor. This CNN predictor was designed to enhance the precision of the prediction errors, thereby facilitating a more efficient data embedding process with the classical prediction error expansion (PEE) embedding strategy. Zhou et al. [18] presented a new RDH method that seamlessly integrates a transformer predictor with an improved PEO-based adaptive embedding strategy. This method, characterized by its multiple embedding rules, is adept at significantly diminishing embedding distortions and elevating the visual quality of the resultant marked images.

From the above discussion, in order to improve performance, existing methods conduct pixel prediction by leveraging adjacent pixels, but these methods [14,15,16,17] ignore the complexity of each pixel with deep learning, which limits the performance of RDH. To tackle the above limitation, in this letter, we improve the CNNP presented in [15] by adding a complexity prediction part to precisely predict pixels’ complexities, which is called the improved CNNP (ICNNP) in the rest of this letter. Specifically, during data embedding, we first split a grayscale image into two sub-images, where one sub-image is predicted by another one. Then, we sort the prediction errors of the predicted pixels based on their complexities, and the prediction errors with less complexity are used for data embedding with a classical HS strategy. Finally, experimental results show that the achieved performance of the proposed ICNNP-based scheme is better than that of the CNNP-based scheme presented in [15].

## 2. Proposed Improved Scheme

### 2.1. Network Architecture

As shown in Figure 1, according to the checkerboard context model [20], we split the cover image into two sub-images, which consist of “Circle” and “Square” pixels, respectively. For the “Circle” sub-image, the values of the “Circle” pixels are maintained, while those of the “Square” pixels are set to 0. Meanwhile, just the values of the “Circle” pixels are set to 0 for the “Square” sub-image. Due to the pixel correlation of the two sub-images, each sub-image is used to predict the pixel values and complexities of the other sub-image.

Figure 2 depicts the architecture of the proposed ICNNP. The architecture of the ICNNP consists of three parts, i.e., feature extraction, pixel prediction, and complexity prediction. “Square” sub-image *I*_2_ is fed to the network to predict the values and complexities of the “Circle” pixels, where the values of complexity are adjusted to [0, 255] for better visualization. The lower the value, the lower the complexity. The feature extraction consists of some convolution layers with different filter sizes (3 × 3, 5 × 5, 7 × 7, ⋯), which are parallelized and appended with a 3 × 3 convolution layer, respectively, to extract features from different receptive fields. A residual block is then applied to further aggregate and refine the learnt features from different branches. With the extracted feature, the pixel prediction yields the predicted “Circle” sub-image ${\tilde{I}}_{1}$, and the complexity prediction yields the predicted complexity ${\tilde{C}}_{1}$ of the “Circle” sub-image *I*_1_. “Conv” stands for the convolution unit with kernel size *S* × *S*, and the number of channels is output × input. A LeakyReLU activation function [33] is located between each two convolution layers.

It is worthy to note that the complexity prediction is similar to the pixel prediction, i.e., instead of orthogonal adjacent pixels [15,20], more adjacent pixels are used to nonlinearly predict the complexity of the pixel area, improving the performance of RDH.

### 2.2. Training

In the ICNNP, the well-trained parameters of the CNNP [15] are loaded into the feature extraction and pixel prediction modules. Note that these parameters are fixed, and the complexity prediction parameters are updated during the training of the ICNNP. During the training, the input of the ICNNP is the “Square” sub-image *I*_2_, while the outputs are the predicted “Circle” sub-image ${\tilde{I}}_{1}$ and the predicted complexity ${\tilde{C}}_{1}$ of the “Circle” sub-image *I*_1_. Since the filter parameters of the feature extraction and the pixel prediction are fixed, the training objective is no longer the “Circle” sub-image *I*_1_ but the referenced complexity $C_{1}$ of *I*_1_. The definition of $C_{1}$ is described as follows:(1)For “Square” pixels, $C_{1}\left(i,j\right)$ is set to 0.(2)For “Circle” pixels, if i=1, i=M, j=1, or j=N, $C_{1}\left(i,j\right)$ is set to 0; otherwise, $C_{1}\left(i,j\right)(2\leq i\leq M-1,2\leq j\leq N-1)$ is calculated as(1)$$C_{1}\left(i,j\right)=\frac{1}{R}\cdot \sqrt{\sum_{k=k_{1}}^{k_{2}}\sum_{l=l_{1}}^{l_{2}}{\left(I\left(i+k,j+l\right)-I\left(i,j\right)\right)}^{2}},$$
where(2)$$\left\{\begin{array}{c} k_{1}=-1,k_{2}=2\;,\;i=2\;\\ k_{1}=-2,k_{2}=1\;,\;i=M-1\;\\ k_{1}=-2,k_{2}=-2\;,\;2<i<M-1 \end{array}\right.,$$(3)$$\left\{\begin{array}{c} l_{1}=-1,l_{2}=2\;,\;j=2\;\\ l_{1}=-2,l_{2}=1\;,\;j=N-1\;\\ l_{1}=-2,l_{2}=-2\;,\;2<j<N-1 \end{array}\right.,$$(4)$$R=\left(k_{2}-k_{1}+1\right)\times \left(l_{2}-l_{1}+1\right)-1.$$


In the proposed scheme, the max predicted pixel area is 5 × 5, which can accurately calculate the pixel complexity.

As with the CNNP in [15], we leverage back-propagation [34] and the Adam algorithm [35] to optimize the objective function defined below:(5)$$Loss=\frac{1}{N}\sum_{i=1}^{N}{\left({\tilde{C}}_{1,i}-C_{1,i}\right)}^{2}+\lambda {\left\|\omega \right\|}_{2}^{2},$$
where *N* is the number of training images, and ${\tilde{C}}_{1,i}$ and $C_{1,i}$ represent the predicted and referenced complexities of the “Circle” sub-image in the *i*-th training image, respectively. *ω* stands for all weights of the network, and λ denotes the weight decay.

### 2.3. Data Embedding of ICNNP-Based RDH

Figure 3 depicts the data embedding architecture of the ICNNP-based RDH scheme. The adopted double embedding strategy [20] with the HS technique [7] involves the successive usage of the “Circle” sub-image embedding and the “Square” sub-image embedding, and the “Square” sub-image embedding is performed after the “Circle” sub-image embedding.

Cover image *I* is firstly separated into two sub-images, i.e., a “Circle” sub-image *I*_1_ and a “Square” sub-image *I*_2_. Next, the predicted “Circle” sub-image ${\tilde{I}}_{1}$ and the predicted complexity ${\tilde{C}}_{1}$ of *I*_1_ are predicted with *I*_2_ as follows:(6)$$\left\{{\tilde{I}}_{1},{\tilde{C}}_{1}\right\}=ICNNP\left(I_{2}\right).$$

Then, the prediction errors of *I*_1_ are calculated as(7)$$e_{1}\left(i,j\right)=I_{1}\left(i,j\right)-{\tilde{I}}_{1}\left(i,j\right),\;\left(i+j\right)mod2\equiv 0,$$
where “≡” represents modular congruence. According to the magnitude of predicted complexities and the size of the additional data *S*_1_, we select the predicted errors with less complexity and determine two thresholds *T_n_*_1_ (*T_n_*_1_ < 0) and *T_p_*_1_ (*T_p_*_1_ ≥ 0) for HS-based data embedding, which is achieved as(8)$$E_{1}\left(i,j\right)=\left\{\begin{array}{c} 2e_{1}\left(i,j\right)+b\;,\;if\;e_{1}\left(i,j\right)\in [T_{n1},T_{p1}]\;\\ e_{1}\left(i,j\right)+T_{p1}+1\;,\;if\;e_{1}\left(i,j\right)>T_{p1}\;\\ e_{1}\left(i,j\right)+T_{n1}\;,\;if\;e_{1}\left(i,j\right)<T_{n1}\; \end{array}\right.,$$
where $b\in \left[0,\;1\right]$ is the data to be embedded, including the encrypted additional data and some auxiliary data [20]. Therefore, the marked “Circle” sub-image *MI*_1_ is generated as(9)$${MI}_{1}\left(i,j\right)={\tilde{I}}_{1}\left(i,j\right)+E_{1}\left(i,j\right).$$

During the “Square” embedding process, due to the requirement that the pattern of the input data must match the “Square” sub-image *I*_2_, the marked “Circle” sub-image *MI*_1_ cannot be directly fed into the network to predict the “Square” sub-image *I*_2_. As illustrated in Figure 4, if the image’s height is even, rotating it clockwise by 90 degrees results in the image $MI_{1}^{\prime }$ matching the pattern of the ’Square’ sub-image *I*_2_. With the same ICNNP, as shown in Equation (10), we feed the rotated marked “Circle” sub-image $MI_{1}^{\prime }$ into the network, and then we obtain the predicted rotated “Square” sub-image ${\tilde{I}}_{2}^{\prime }$ and its complexity ${\tilde{C}}_{2}^{\prime }$.(10)$$\left\{{\tilde{I}}_{2}^{\prime },{\tilde{C}}_{2}^{\prime }\right\}=ICNNP\left(MI_{1}^{\prime }\right).$$

Then, ${\tilde{I}}_{2}^{\prime }$ and ${\tilde{C}}_{2}^{\prime }$ are rotated counterclockwise by 90 degrees to obtain the predicted “Square” sub-image ${\tilde{I}}_{2}$ and the predicted complexity ${\tilde{C}}_{2}$ of *I*_2_. Similarly to the “Circle” embedding, another part of additional data *S*_2_ is encrypted with *K* and then embedded into *I*_2_ to obtain the marked “Square” sub-image *MI*_2_.

Finally, we combine the marked “Circle” sub-image *MI*_1_ and the marked “Square” sub-image *MI*_2_ to obtain the marked image *MI*.

### 2.4. Extraction and Image Recovery of ICNNP-Based RDH

Figure 5 describes the architecture of data extraction and image recovery, which are the reverse procedures of data embedding, so we perform the “Circle” extraction/recovery ahead of the “Square” extraction/recovery. The marked image *MI* is firstly divided into two sub-images, i.e., the marked “Circle” sub-image *MI*_1_ and the marked “Square” sub-image *MI*_2_. With the rotated marked “Circle” sub-image $MI_{1}^{\prime }$, ${\tilde{I}}_{2}^{\prime }$ and ${\tilde{C}}_{2}^{\prime }$ are predicted using the ICNNP as in Equation (10). ${\tilde{I}}_{2}$ and ${\tilde{C}}_{2}$ are then obtained by rotating ${\tilde{I}}_{2}^{\prime }$ and ${\tilde{C}}_{2}^{\prime }$ respectively. Next, the marked prediction errors of *I*_2_ are calculated as(11)$$E_{2}\left(i,j\right)={MI}_{2}\left(i,j\right)-{\tilde{I}}_{2}\left(i,j\right),\;\left(i+j\right)mod2\equiv 1.$$
According to the sorted magnitude of ${\tilde{C}}_{2}$ and the extracted auxiliary data *T_n_*_2_ (*T_n_*_2_ < 0) and *T_p_*_2_ (*T_p_*_2_ ≥ 0) as thresholds, the data extraction is operated as(12)$$b=E_{2}\left(i,j\right)\;mod\;2\;,\;E_{2}\left(i,j\right)\in [2T_{n2},2T_{p2}+1],$$
and the original prediction errors of *I*_2_ are recovered as(13)$$e_{2}\left(i,j\right)=\left\{\begin{array}{c} \left\lfloor E_{2}\left(i,j\right)/2\right\rfloor \;,if\;E_{2}\left(i,j\right)\in [2T_{n2},2T_{p2}+1]\;\\ E_{2}\left(i,j\right)-T_{p2}-1,if\;E_{2}\left(i,j\right)>2T_{p2}+1\;\\ E_{2}\left(i,j\right)-T_{n2}\;,if\;E_{2}\left(i,j\right)<2T_{n2}\; \end{array}\right.,$$
where $\left\lfloor \cdot \right\rfloor$ is the floor function. We decrypt the extracted bits to obtain *S*_1_ with *K*, and recover the cover “Square” sub-image *I*_2_ as(14)$$I_{2}\left(i,j\right)=e_{2}\left(i,j\right)+{\tilde{I}}_{2}\left(i,j\right),\;\left(i+j\right)mod2\equiv 1$$

Similarly, *S*_1_ is extracted correctly, and the cover “Circle” sub-image *I*_1_ is recovered losslessly. Finally, we combine the recovered “Square” sub-image *I*_2_ and “Circle” sub-image *I*_1_ to obtain *I*.

## 3. Experimental Results

To assess the efficiency of the proposed ICNNP-based scheme, the parameters of its complexity prediction model were trained with 1000 grayscale images of size 512 × 512, which were selected from BOWS-2 [36] randomly. The ICNNP was trained on an Intel Core i10 CPU (3.6 GHz) with 16 GB of RAM and an NVIDIA GeForce RTX 2060. The equipment was sourced from Lenovo, located in Beijing, China. The weight decay λ was 1 × 10^−3^, the batch size was set to 4, and the initial learning rate was set to 1 × 10^−3^. The number of training epochs was 20, and the optimizer used for training the ICNNP was Adam. In [15], the prediction performance of the CNNP is proved to be better than some traditional linear predictors, such as MEDP, GAP, etc., and the achieved rate-distortion performance is better than that of the above predictor with the expansion embedding scheme. It is also better than that of BIP with the HS scheme, and the performance of HS is far better than that of expansion embedding with the CNNP. Therefore, we just evaluate the performance with the ICNNP by comparing it with that of the CNNP [15] with the same HS technique.

With the four 512 × 512 grayscale images shown in Figure 6 as cover images with the same embedding capacity (EC), we employed a PSNR (peak signal-to-noise ratio) between the marked image and the original image as the metric for objective image quality evaluation. In addition, we chose 100 grayscale images of size 512 × 512 randomly that were different from the training images from BOWS-2 [36], and we tested them with different embedding capacities (ECs) to evaluate the universality of the ICNNP.

Figure 7 shows the PSNR values of four test images with different Ecs, which indicates that the PSNRs of the ICNNP-based RDH scheme are larger than those of the CNNP-based RDH scheme. Figure 8 shows the experimental results when embedding 10,000 bits of data, corresponding to an embedding rate of approximately 0.038 bpp. The PSNRs between the stego images and the original images are 57.02 dB, 60.73 dB, 51.81 dB, and 56.60 dB, respectively. The stego images are visually indistinguishable from the original images shown in Figure 6. From Table 1, we can see the average PSNRs of 100 test images for different ECs. When the EC is 10,000 bits, the mean PSNR achieved by the ICNNP-based RDH scheme is 62.37 dB, while that of the CNNP-based RDH scheme is 61.31 dB, which is 1.06 dB lower than that of the proposed scheme. Along with the EC increases from 20,000 bits to 150,000 bits, the mean PSNRs of the ICNNP-based RDH scheme are 0.81 dB, 0.60 dB, 0.55 dB, 0.56 dB, 0.55 dB, 0.54 dB, 0.53 dB, 0.51 dB, 0.47 dB, 0.41 dB, 0.37 dB, 0.33 dB, 0.26 dB, and 0.20 dB higher than those of the CNNP-based RDH scheme with the same EC, respectively.

## 4. Conclusions

The improved CNN predictor for RDH presented in this paper extracts features from different receptive fields with whole optimization and uses more neighboring pixels to precisely predict the pixel value and its complexity. During data embedding, a grayscale image is split into two sub-images, one sub-image is applied to predict the other sub-image alternately using the ICNNP. Then, the pixels’ prediction errors are sorted according to the pixels’ prediction complexities, and the prediction errors with less complexity are chosen for data embedding with the classical HS strategy. The original image is recovered losslessly, and the embedded data are extracted correctly. The data extraction and image recovery are separable. Experimental results demonstrate that the proposed ICNNP, when combined with the classical histogram shifting (HS) strategy, achieves superior performance compared to the CNNP presented in [15] using the same HS strategy, thereby proving its effectiveness.

The proposed ICNNP is the first to predict pixel errors and their complexity for RDH. Employing joint training for pixel prediction and complexity prediction holds the promise of achieving superior RDH performance. In future work, we plan to contemplate the joint optimization of pixel error prediction and pixel complexity prediction using advanced deep learning methods, as well as the integration of various embedding strategies, such as PEE, PEO, and multi-histogram shifting (MHS).

## Figures and Tables

**Figure 1 entropy-27-00159-f001:**
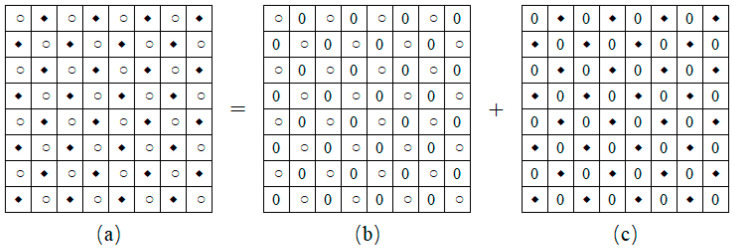
Illustration of splitting an original image into two sub-images. (**a**) Original image *I*. (**b**) “Circle” sub-image *I*_1_. (**c**) “Square” sub-image *I*_2_.

**Figure 2 entropy-27-00159-f002:**
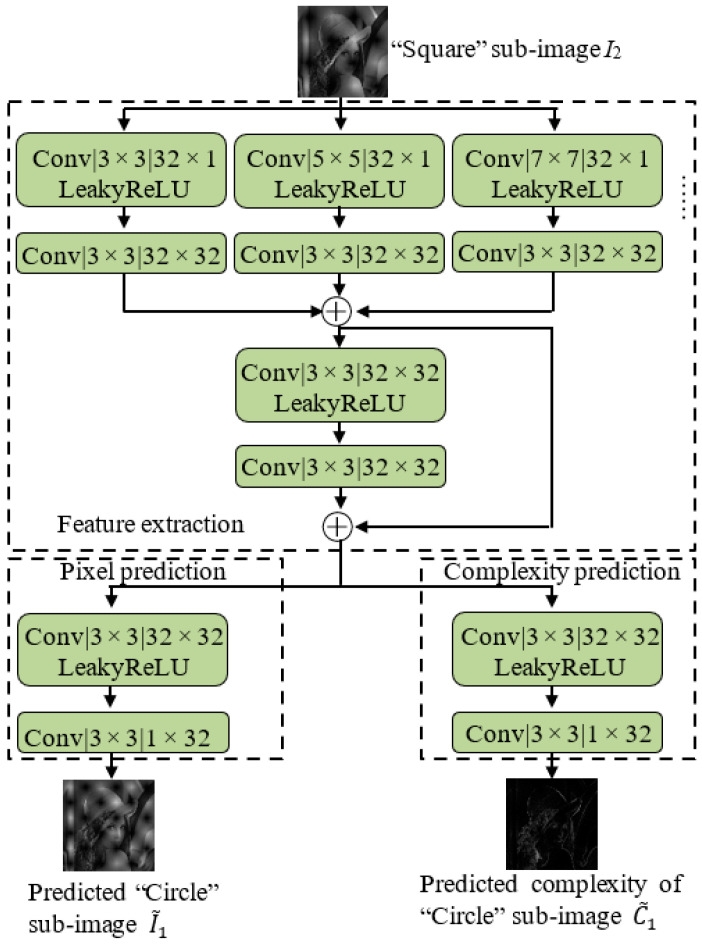
The overall architecture of the proposed ICNNP.

**Figure 3 entropy-27-00159-f003:**
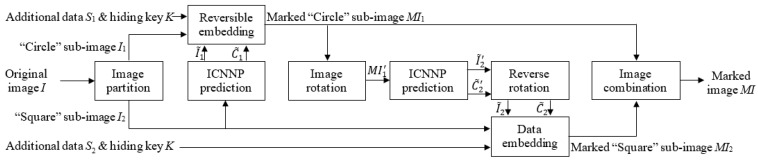
The flowchart of data embedding using the ICNNP-based RDH scheme.

**Figure 4 entropy-27-00159-f004:**
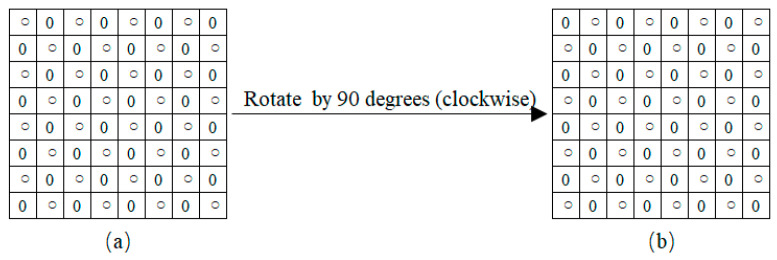
Illustration of image rotation of marked “Circle” image. (**a**) ${MI}_{1}$. (**b**) $MI_{1}^{\prime }$.

**Figure 5 entropy-27-00159-f005:**
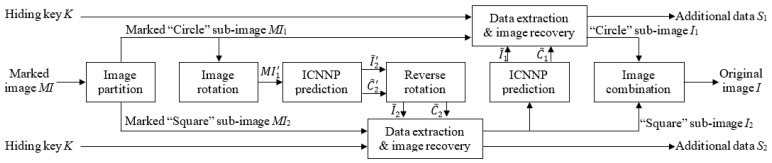
The flowchart of data extraction and image recovery using the ICNNP-based RDH scheme.

**Figure 6 entropy-27-00159-f006:**
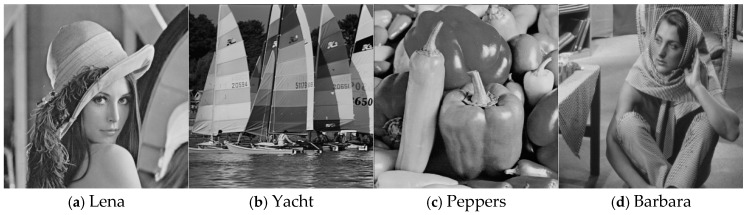
Four cover images.

**Figure 7 entropy-27-00159-f007:**
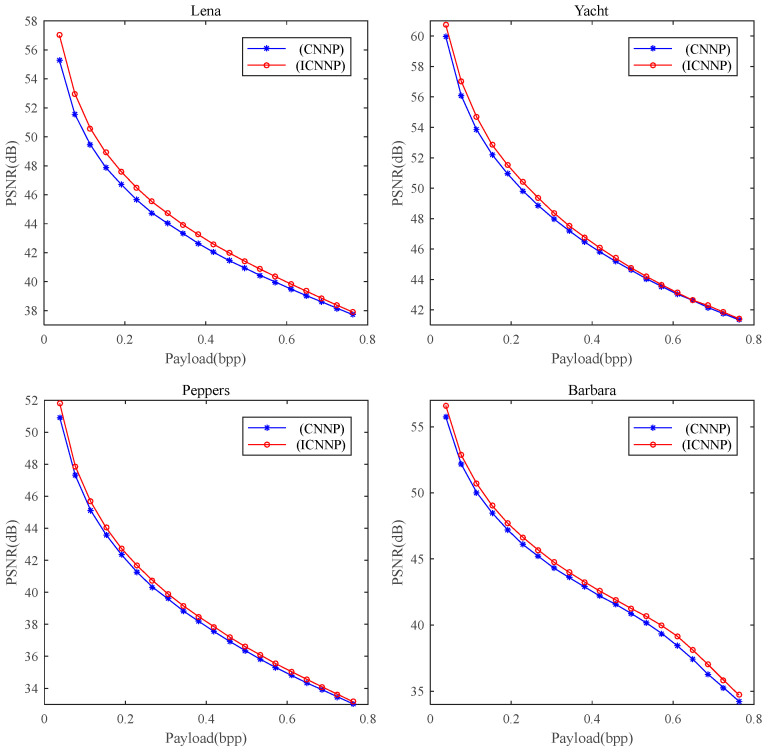
Performance comparison of CNNP in [15] and ICNNP for RDH on four test images.

**Figure 8 entropy-27-00159-f008:**
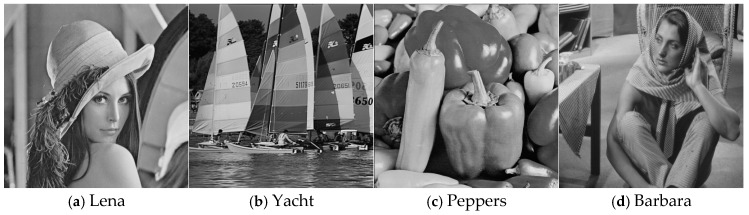
Some experimental results.

**Table 1 entropy-27-00159-t001:** Average PSNR (dB) values on 100 test images of ICNNP-based RDH method and CNNP-based RDH method [15].

Embedding Capacity (Bits)	Embedding Rate (bpp)	CNNP [15]	ICNNP
10,000	0.038	61.31	62.37
20,000	0.076	58.01	58.82
30,000	0.114	55.98	56.58
40,000	0.153	54.43	54.98
50,000	0.191	53.10	53.66
60,000	0.229	51.94	52.49
70,000	0.267	50.86	51.40
80,000	0.305	49.85	50.38
90,000	0.343	48.88	49.39
100,000	0.381	47.91	48.38
110,000	0.420	46.95	47.36
120,000	0.458	46.03	46.40
130,000	0.496	45.13	45.46
140,000	0.534	44.29	44.55
150,000	0.572	43.47	43.67

## Data Availability

The data presented in this study are available on request from the corresponding author.

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
