# Peer review of "Improved CNN Prediction Based Reversible Data Hiding for Images"

_entropy, 2025, doi:10.3390/e27020159_

Round 1

Reviewer 1 Report

Comments and Suggestions for Authors

In this paper, the authors propose a reversible data hiding (RDH) method for images with an improved convolutional neural network predictor (ICNNP). This paper introduces an innovative complexity prediction module that not only enhances the accuracy of pixel value prediction but also enables the precise simultaneous prediction of pixel complexity. With this improvement, prediction errors of low complexity can be selectively chosen to achieve low-distortion data embedding. The experimental results show better embedding performance. This work is technically sound, and this paper can be accepted after minor revisions, as outlined below.

1. On Line 91–Line 92, the phrase “for a good visualization display” should be modified to “for better visualization”.

2. The symbol “ ”, which represents the predicted “Circle” sub-image, is not shown in Figure 2.

3. Equation (5) needs a more precise explanation of the variable “i” .

4. On Line 145, the phrase “two threshold” should be modified to “two thresholds”.

5. In Equations (7), (11), and (14), the meaning of the symbol “≡” has not been defined. It is better to clarify its meaning where it first appears.

Author Response

Comments 1: On Line 91–Line 92, the phrase “for a good visualization display” should be modified to “for better visualization”.

Response 1:Thank you for pointing this out. We agree with this comment. Therefore, we have modified the phrase “for a good visualization display” to “for better visualization”. This change can be found on lines 91–92 of the revised manuscript. Changes are highlighted in red for ease of review.

Comments 2: The symbol “ ”, which represents the predicted “Circle” sub-image, is not shown in Figure 2.

Response 2:Thank you for pointing this out. We agree with this comment. Therefore, we have added the symbol “” in Figure 2. This change can be found in Figure 2 of the revised manuscript.

Comments 3: Equation (5) needs a more precise explanation of the variable “i” .

Response 3:Thank you for pointing this out. We agree with this comment. Therefore, we have modified Equations (5), and added,  represent the predicted and referenced complexities of the ‘Circle’ sub-image in the i-th training image, respectively” to make it clearer. This change can be found on lines 128-129 of the revised manuscript. Changes are highlighted in red for ease of review.

Comments 4: On Line 145, the phrase “two thresholds” should be modified to “two thresholds”.

Response 4:Thank you for pointing this out. We agree with this comment. Therefore, we have modified the phrase “two threshold” to “two thresholds”. This change can be found on line 147 of the revised manuscript. Changes are highlighted in red for ease of review.

Comments 5: In Equations (7), (11), and (14), the meaning of the symbol “≡” has not been defined. It is better to clarify its meaning where it first appears.

Response 5:Thank you for pointing this out. We agree with this comment. Therefore, we have added “where, ‘≡’ is represents modular congruence.” when the symbol “≡” first appears in Equations (7). This change can be found on line 145 of the revised manuscript. Changes are highlighted in red for ease of review.

Reviewer 2 Report

Comments and Suggestions for Authors

The manuscript introduces a novel RDH technique based on an improved CNN prediction (ICNNP), which is composed of three modules: feature extraction, pixel prediction, and complexity prediction. With the addition of a complexity prediction module, both pixel values and their complexities can be accurately predicted during the embedding process. As a result, prediction errors in low-complexity regions can be selectively exploited to achieve data embedding with minimal distortion. This work is technically sound, and the experimental results demonstrate the effectiveness of the method. In my opinion, this manuscript can be accepted after a minor revision. The suggestions are listed as follows.

1. In the data embedding process of the ICNNP-based RDH, the authors propose an image rotation strategy. However, the description of this strategy is insufficient. Therefore, the authors should provide a clear and detailed explanation of the image rotation process.

2. In Figure 3 and Figure 5, it would be better to modify “Results rotation” to “Reverse rotation”.

3. The sizes of the four test images and the 100 images chosen from BOWS-2 should be mentioned.

Author Response

Comments 1: In the data embedding process of the ICNNP-based RDH, the authors propose an image rotation strategy. However, the description of this strategy is insufficient. Therefore, the authors should provide a clear and detailed explanation of the image rotation process.

Response 1: Thank you for pointing this out. We agree with this comment. Therefore, we have made efforts to provide a clearer explanation of the image rotation process. This change can be found on lines 153-157 of the revised manuscript. Changes are highlighted in red for ease of review.

Comments 2: In Figure 3 and Figure 5, it would be better to modify “Results rotation” to “Reverse rotation”.

Response 2: Thank you for pointing this out. We agree with this comment. Therefore, we have modified “Results rotation” to “Reverse rotation” in Figure 3 and Figure 5. This change can be found in Figure 3 and Figure 5 of the revised manuscript.

Comments 3: The sizes of the four test images and the 100 images chosen from BOWS-2 should be mentioned.

Response 3: Thank you for pointing this out. We agree with this comment. Therefore, we have provided the sizes of the four test images and the 100 images chosen from BOWS-2. This change can be found on lines 194, 205, and 208 of the revised manuscript. Changes are highlighted in red for ease of review.

Reviewer 3 Report

Comments and Suggestions for Authors

This paper proposes an improved convolutional neural network predictor (ICNNP) for reversible data hiding (RDH), introducing a complexity prediction module to enhance pixel complexity estimation. Combined with histogram-shifting (HS), the method outperforms traditional CNNP-based RDH in rate-distortion and image quality. Experimental results demonstrate its effectiveness in preserving visual quality and achieving high embedding capacity. While the paper demonstrates notable contributions to the field, several aspects need minor revisions to enhance clarity and completeness. These points are summarized as follows:

  1. The paper does not specify the dimensions or resolution of the test images used in the experiments.

  2. The embedding capacity is currently reported in absolute terms (bits). For better comparability and standardization, the authors should also express the embedding capacity in bits per pixel (bpp).

  3. The paper does not include information about the number of training epochs or the specific optimizer used for training both ICNNP and CNNP. Including these details would improve the reproducibility of the results.

  4. No figures are provided to visually demonstrate the quality of the stego images generated by the proposed method.

Author Response

Comments 1: The paper does not specify the dimensions or resolution of the test images used in the experiments.

Response 1: Thank you for pointing this out. We agree with this comment. Therefore, we have specified the dimensions of the test images used in the experiments. This change can be found on lines 194, 205, and 208 of the revised manuscript. Changes are highlighted in red for ease of review.

Comments 2: The embedding capacity is currently reported in absolute terms (bits). For better comparability and standardization, the authors should also express the embedding capacity in bits per pixel (bpp).

Response 2: Thank you for pointing this out. We agree with this comment. Therefore, we have expressed the embedding capacity in bits per pixel (bpp). This change can be found in Figure 7 and Table 1. Changes in Table 1 are highlighted in red for ease of review.

Comments 3: The paper does not include information about the number of training epochs or the specific optimizer used for training both ICNNP and CNNP. Including these details would improve the reproducibility of the results.

Response 3: Thank you for pointing this out. We agree with this comment. Therefore, we have added  the information about the number of training epochs and the specific optimizer used for training ICNNP. This change can be found on line 198. Changes are highlighted in red for ease of review.

Comments 4: No figures are provided to visually demonstrate the quality of the stego images generated by the proposed method.

Response 4: Thank you for pointing this out. We agree with this comment. Therefore, we have added Figure 8 to visually demonstrate the quality of the stego images generated by the proposed method. This change can be found in Figure 8 of the revised manuscript. The explanation of Figure 8 can be found in lines 219-221 and 228-229. Changes are highlighted in red for ease of review.